



# A method for the spectral analysis and identification of Fog, Haze and Dust storm using MODIS data

**Qinghua Su[1, 2], Lin Sun[1], Mei Di[1], xinyan Liu[1], Yikun Yang[1]**

[1] Geomatics College, Shandong University of Science and Technology, Shandong Qingdao 266590, China
[2] School of Geography and Tourism, Qufu Normal University, Qufu 273165, China

Correspondence to: Lin Sun (sunlin6@126.com)

**Abstract.** The three typical extreme weather of fog, haze and dust storm have occurred frequently in recent years in China. These events influence the transportation, the ecological environment, and the daily lives of people. Remote sensing is very important technology that can be used to monitor them due to its high temporal resolution and wide area of coverage. But because the spectral features of the three extreme weather conditions are very complex, the high accuracy identification of them is facing severe challenges. In this article, the spectra of these three weather conditions, as well as those of clouds and the background surface, are analyzed. A monitoring model is constructed to achieve the separation of fog, haze, dust storm, clouds and the underlying surface using satellite data. The monitoring results are tested based on their corresponding measurements obtained from ground stations, and indicate that the extraction of fog, haze and dust storm can reach a high accuracy .

## 1 Introduction

Fog, Haze and Dust storm are three typical extreme weather that often occur in China, especially in the areas where the ecological environment is seriously damaged. (Miri et al., 2010; Zhang et al., 2013; Sallis et al., 2014; David et al., 2012; Naksen et al., 2017). Fog, haze and dust storm can cause the climate changes by affecting the radiation budget and energy balance of the Earth-atmosphere system (Quinn and Bates, 2003; Kaskaoutis et al., 2006; Elias et al., 2009; Maghrabi, 2017). When such weather happens, atmospheric visibility is sharply reduced, which seriously affects the transportation (Sisler and Malm, 1994; Deng et al., 2011). Fog,haze and dust particles can also produce air pollution (Yu et al., 2011; Kang et al. 2013; Quan et al., 2014; Gao and Chen, 2016), increasing the concentration of inhalable particles and leading to respiratory and cardiovascular diseases (Thurston et al., 1994; Bai et al., 2006; Zhao et al., 2013; Gao et al., 2016; Gao et al., 2017).



The methods used to detect typical extreme weather mainly include ground observations and remote sensing detection. Because the station is less and the spatial distribution is uneven, it is difficult to truly reflect the spatial distribution of the extreme weather with the method of ground observations(Zhang et al., 2005). Remote sensing technology, which have features of high frequency and wide coverage, can play a more important role in the

detection and quantitative analysis of spatial and temporal distribution characteristics of them (Ellrod, 1995; Ackerman, 1997; Qu et al., 2006; Huang et al., 2007; Lee and Kim, 2010). However, it is very difficult to distinguish the fog, haze and dust precisely because of the similarity of the spectral and textural features of them, and also they were influenced by clouds, which also have the similar spectral and textural properties to them. Thus, it is urgent to study the detection methods of fog, haze and dust storm to reduce the limitations of ground stations and

decrease the losses that are caused by typical extreme weather.

The remote sensing detection of fog began in 1970s, meteorological satellites, such as NOAA/AVHRR, GOES and MODIS, have been widely used to do such work. For example, Eyre et al. used NOAA/AVHRR data to distinguish fog and low clouds, difference of the brightness temperatures between channel 3 (3.7 μm) and channel 4 (11.0 μm) was selected as a basis for discrimination (Eyre et al., 1984). Bendix et al. proposed a fog detection method to detect

the daytime and nighttime fog distribution with combining use of the Terra-MODIS and MSG-SEVIRI data(Bendix et al., 2004).

As one of the main atmospheric pollutants, haze has attracted much attention, as it has occurred frequently in recent years. Studies of haze have mainly focused on its spatial and temporal distribution, source, transmission and climatic impact. Malm quantitatively analyzed the spatial and temporal dynamic change of haze distribution in American

continental, and so to track and simulate the origin of the haze materials (Malm, 1992). Okada et al. analyzed the mixture state of individual aerosol particles in the 1997 Indonesian haze episode (Okada et al., 2001).

In the 1970s, detection of dust storm with remote sensing technology developed rapidly. Shenk and Curran first began to study the detection methods of dust storm using visible and infrared channels (Shenk and Curran, 1975). Ackerman identified the dust storm using radiative temperature difference at 3.7 and 11 μm (Ackerman, 1989).

Roskovensky and Liou established a discrimination function based on the reflectance ratio of visible light and the brightness temperature difference of the thermal infrared channel to distinguish cirrus clouds and dust storm (Roskovensky and Liou, 2005).

Detecting the fog, haze and dust storm still faces great challenge for the following reasons: (1) It is difficult to distinguish between low clouds and fog, as they have similar spectral features. (2) The remote sensing detection of haze is rarely performed, current haze detection still mainly depended on the ground observation data. (3) Previous studies did not distinguish the three extreme weather at the condition of simultaneous occurrence of them. From a

wide range of monitoring, such situation often occurs, but because of their optical characteristics are very similar, monitoring in this case is very difficult.

The physical characteristics of fog, haze and dust storm are much different, they have many differences in particle size, moisture content and so they show different spectral characteristics, especially in the visible and infrared wavelength. Remote sensing technology takes advantage of this optical difference to realize the identification of

them. MODIS (Moderate Resolution Imaging Spectrometer) data are used to do the experiment for the detection of the three extreme weather phenomena. Based on the spectral analysis of this three kinds of extreme weather and the remote sensing data characteristics, this paper construct a high accuracy model for the fog, haze and dust storm identification.

## 2 Detection model construction based on MODIS data

### 15   2.1 MODIS data and processing

MODIS is a key instrument aboard the Terra and Aqua satellites. They are acquiring data in 36 spectral bands, ranging in wavelength from 0.4 μm to 14.4 μm. Two bands of 0.66 μm and 0.86 μm are imaged at a nominal resolution of 250 m at nadir, with five bands of visible to near infrared band at 500 m, and the remaining 29 bands at 1 km, they achieves a 2,330km swath and provides global coverage every one to two days. There are many standard

MODIS data products that scientists are using to study global change. These products are being used by scientists from a variety of disciplines, they are playing an important role in ecological environment monitoring and monitoring of global climate change.

### 2. 2 Data preprocessing

The Level 1B data set contains calibrated and geolocated radiances generated from MODIS Level 1A sensor counts.

The radiances are in W/(m2- μm-sr). The parameters of apparent reflectance and brightness temperature are used to construct the detection model. Apparent reflectance is the ratio of upwelling irradiance to the downwelling irradiance,



and can be calculated for the solar reflective bands through knowledge of the solar irradiance with the following

equation:

$$ref = \frac{\pi L D^2}{E \cos \theta} \qquad (1)$$

where ref represents the apparent reflectance, E is the solar irradiance, and D is the earth-sun distance, which is

generally defined as 1. $\cos \theta$ is the cosine of the solar zenith angle.

The brightness temperature is a measurement of the radiance of the thermal infrared radiation traveling upward

from the top of the atmosphere to the satellite, expressed in units of the temperature of an equivalent black body.

The brightness temperature is the fundamental parameter measured by thermal infrared radiation sensors, which can

be obtained from Planck's formula.

$$L = \frac{2hc^2\lambda^{-5}}{(e^{\frac{hc}{k\lambda T}} - 1)} \qquad (2)$$

$$T = (\frac{hc}{k\lambda})\frac{1}{\ln(2hc^2\lambda^{-5}L^{-1}+1)} = \frac{14.39474\times10^3}{\lambda \times \ln(\frac{119.109\times10^6}{\lambda^5 \times L}+1)} \qquad (3)$$

where h is Planck's constant, which is defined as $6.626\times10^{-34}$ J.s; k is the Boerziman constant, which is defined as

$1.3806\times10^{-23}$ J.K$^{-1}$; c is the velocity of light, which is defined as $2.998\times10^8$ m.s$^{-1}$; $\lambda$ is wavelength, and T is

temperature.

**2.3 Construction the model of fog, haze and dust storm detection**

According to the monitoring results obtained by the meteorological department, three extreme weather at different

time were selected for model building, they are heavy fog weather occurred in regions of Hebei, Henan, Anhui,

Jiangsu, and Shandong on Jan. 27, 2006, haze weather occurred in regions of Shanxi, Hebei, Beijing, Tianjin,

Shandong, and Henan on Dec. 21, 2006, and dust storm weather occurred in the middle-eastern region of Inner

Mongolia on Mar. 9, 2006. Corresponding satellite images are shown in Fig. 1, 2 and 3. Green dots in the images

represent observation stations.



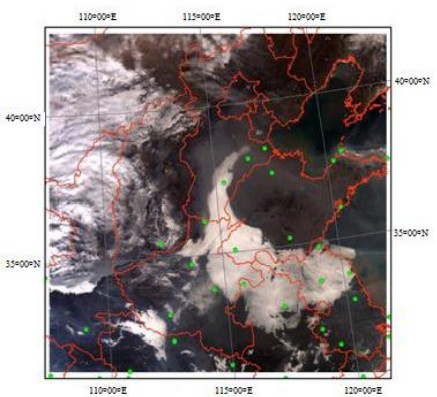

**Fig 1.** The true color image of fog at 10:55 on 01.27.2006

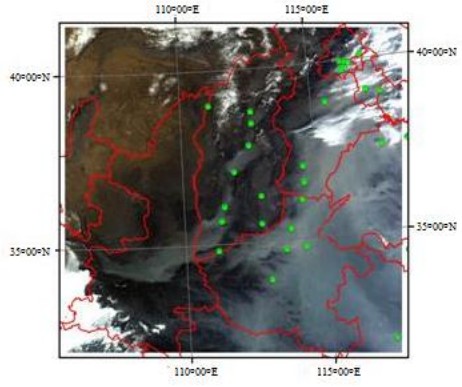

**Fig 2.** The true color image of haze at 11:45 on 12.21.2006

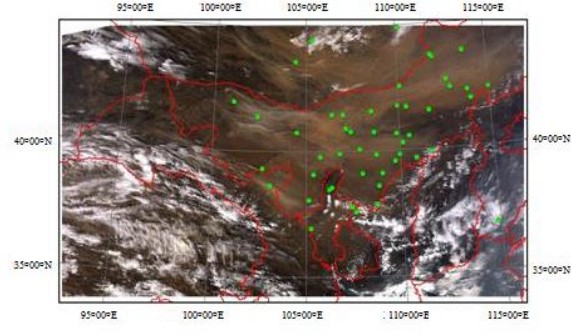

**Fig 3.** The true color image of dust storm at 14:05 on 03.09.2006





Fig. 1 is the true color image of dense fog weather, which presents as white with a smooth top structure, even

texture and clear borders. Fig. 2 is the true color image of haze weather, which presents as gray with an even texture

and clear borders. Fig. 3 is the true color image of dust storm weather, which presents as drab yellow feathers with a

uniform top structure and fuzzy borders. In each image, cloud areas are present as white regions with variable

brightness, rough top structures and messy borders.

Band 1~19 and 26 of the MODIS are visible and near infrared bands, the energy received by these bands is from

the reflected solar radiation of objects, bands 20~25 and 27~36 are thermal infrared bands, the energy received by

the satellite mainly originated from the emitted radiation of the atmosphere and the land. The difference of

reflectance or brightness temperatures of fog, haze, dust storm, clouds and the underlying surface were analyzed

based the sample areas selected from the MODIS images of Fig. 1, 2 and 3. The conclusions of this analysis are as

follows: the combination of bands 22 and 23 can be used to define fog areas, bands 2 and 8 can be used to detect the

haze areas, and bands 31 and 32 can be used to identify dust storm areas.

Fig. 4 is the distribution of the brightness temperatures of fog, haze, dust storm, clouds and the underlying surface

in different image of bands 22, 23, it shows that the differences in cloud brightness temperatures are distributed

widely between 1-11K, peaking at 5K; the differences in the brightness temperatures of the underlying surface, haze

and dust storm are distributed between 3-11K, 3-11K and 4-11K, respectively, and peak at 7K, 6K and 7K; the

differences in fog brightness temperatures are concentrated between 9-12K. Thus, the fog area can be identified if

$T22 - T23 > 9$.

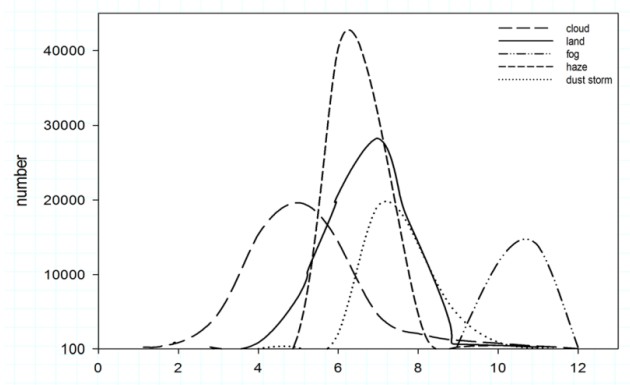

**Fig 4.** The distribution of the brightness temperatures of fog, haze, dust storm, clouds and the underlying surface in difference

image of bands 22,23





Haze has a better scattering intensity in visible bands than it does in infrared bands so the MODIS reflectivity in band 8(b8) is larger than that in band 2(b2). Thus, a Haze Index (HI) was proposed to distinguish between haze and other targets,as shown by following equation:

$$HI = \frac{b8-b2}{b8+b2}.$$ (4)

5    Fig. 5 is the HI of fog, haze, dust storm, clouds and the underlying surface. The HI of the dust storm ranges from -0.42~0.15, peaking at -0.33, HI of the underlying surface reflectance ranges from -0.33~0.12, fog and clouds are distributed from -0.06~0 and -0.03~0.12, respectively. The HI of haze is larger than 0.03; thus, a haze area can be identified if HI >0.03.

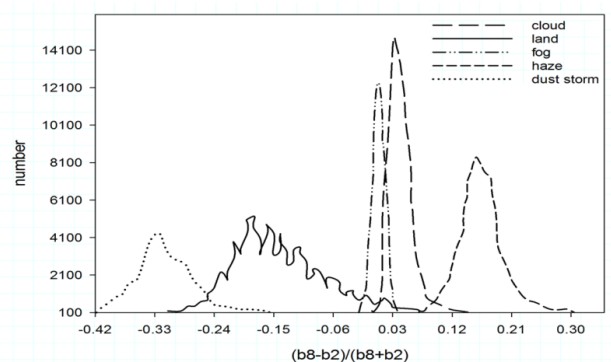

10    **Fig 5.** The HI of fog, haze, dust storm, clouds and the underlying surface in a 2, 8 band combination image

Fig. 6 represents the brightness temperature difference of bands 31 and 32, which is calculated by T31-T32, T31 and T32 are the brightness temperature of band 31 and 32, respectively. The distributions of dust storm, the underlying surface, fog, haze and clouds are -3.7~-1K, -1~0.3K, -0.2~0.6K, -0.1~1K and -0.1~2.3K, respectively, which peak at -2.6K, -0.1K, 0K, 0.1K and 0.7K. The differences in the brightness temperatures of dust storm are

15    quite different than those of other objects at -1K. Thus, a threshold of -1 could be used to identify dust storm areas, which is expressed as $T31-T32<-1$.



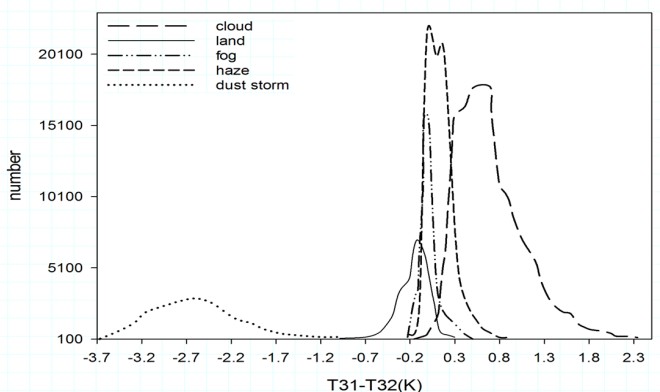

**Fig 6.** The distribution of the brightness temperatures of fog, haze, dust storm, clouds and land in difference image of bands 31, 32

The above combination of wavebands could be used to effectively distinguish between fog, haze and dust storm areas. However, the underlying surface effect must also be eliminated. The combinations of wavebands and bands used to separate the underlying surface are described as follows.

Fig. 7 shows the difference image of bands 20 and 31, in which the green dots represent fog observation stations. Fog areas are expressed in white, cloud areas are expressed in white and black-gray, and the underlying surface is ash-black. In the brightness temperature difference image, differences in color represent differences in brightness temperatures. Fig. 8 indicates that the different distributions of fog, cloud and the underlying surface in bands 20 and 31 are 20-36K, 8-48K and 8-20K, respectively. These distributions of differences in brightness temperatures are consistent with the observed differences in color. A threshold of 20, expressed as $T20 - T31 > 20$, could thus be used to eliminate the underlying surface effect.



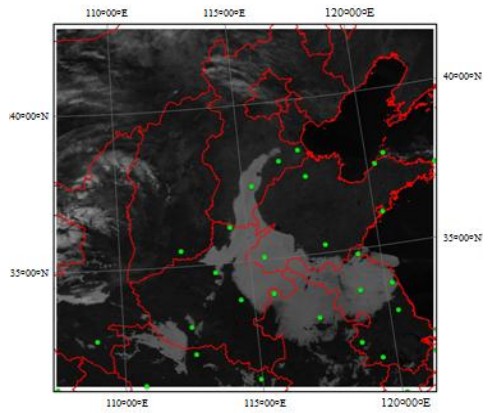

**Fig 7.** The MODIS difference image of bands 20, 31

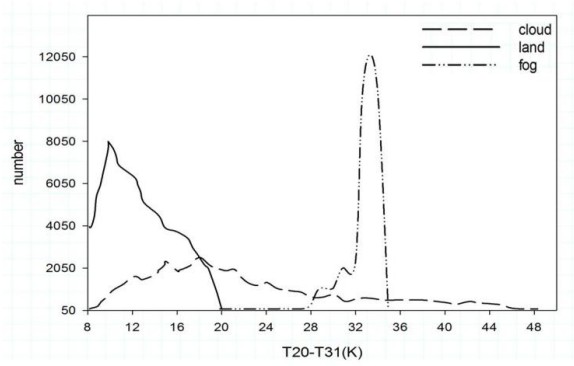

**Fig 8.** The distribution of the brightness temperatures of fog, cloud and land in the difference image of bands 20, 31

5      MODIS band 19 with wavelength ranging from 0.915 to 0.965 μm is an strong moisture absorption waveband, where the reflectance of the underlying surface, clouds (except for high-level clouds) and haze are lower. However, in band 3 with wavelength ranging from 0.459 to 0.479 μm, the reflectance of cloud and haze are higher than that of the underlying surface. So band 19 and band 3 are combined used to identify the underlying surface with the following formula:

10      $UI = (b19-b3)/(b19+b3)$                                          (5)

Fig. 9 shows the UI distribution, where green dots represent haze observation stations. Fig. 10 is the histogram of UI for different objects. It shows that the UI of haze and clouds range from -0.58~-0.1, whereas those of the underlying surface are -0.1~0.26. Defining a threshold of UI<-0.1 could identify the underlying surface.





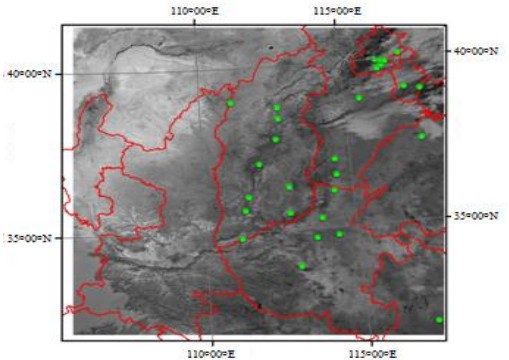

**Fig 9.** The MODIS difference image of bands 19 and 3

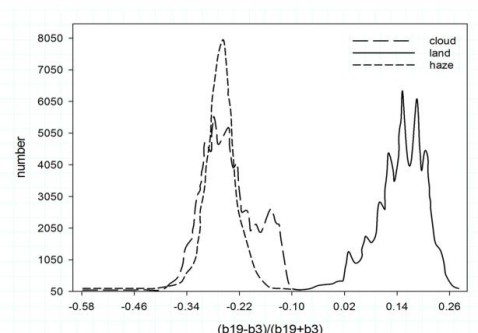

**Fig 10.** The distribution of the reflectance of haze, clouds and land in the 19, 3 band combination image

5        Fig. 11 is the reflectance distribution of the MODIS band 1, the green dots is the dust storm observation stations. It

shows the cloud areas have the highest reflectance, followed by dust storm, the underlying surface is the lowest. Fig.

12 shows that the reflectance of dust storm ranges from 0.12 to 0.3, the underlying surface is 0.06-0.22 and the

clouds is 0.12-0.66. Part of the underlying surface could be identified using a threshold of 0.12.



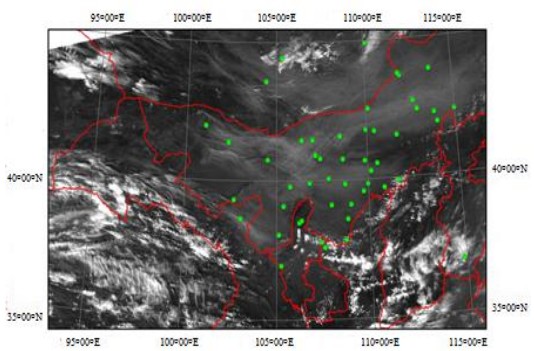

**Fig 11.** The MODIS image of band 1

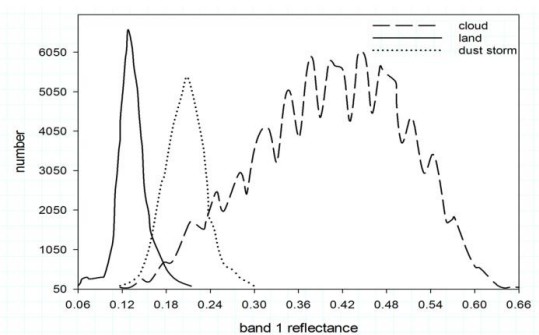

**Fig 12.** The distribution of the reflectance of dust storm, clouds and land in band 1

5       Fig. 13 is the difference image of bands 20 and 31, where the green dots represent ground sites that observed dust

storm. It can be seen that the color of dust storms is similar to that of clouds, and they are much different with that of

the underlying surface. Fig. 14 shows that the brightness temperatures of dust storm, clouds and the underlying

surface. It shows that the brightness temperature of the underlying surface is the lowest, and could be separated using

a threshold of $T20 - T31 > 24$.





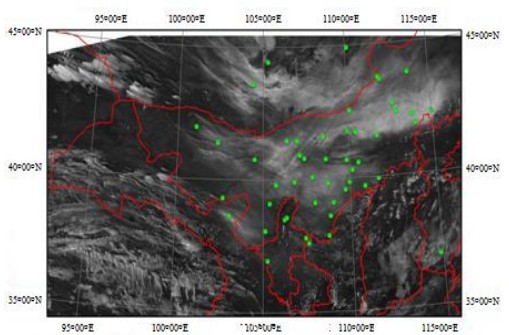

**Fig 13.** The MODIS difference image of bands 20 and 31

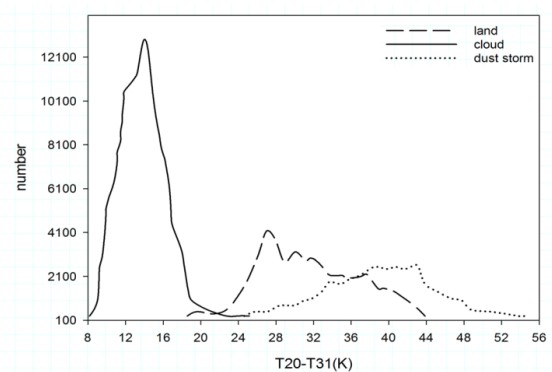

**Fig 14.** The distribution of the brightness temperatures of dust storm, clouds and land in difference image of bands 20 and 31

## 3. Experiment and validation

### 3.1 Validation data introduction

The data used for validating the extreme weather detection method is obtained from MICAPS (Meteorological Information Comprehensive Analysis and Process System), which is a software tool for visualizing data, and is popular used by China Meteorological Administration (CMA) for weather forecast. A total of 19 kinds of data is in the MICAPS, include the ground observation data, satellite data, and so on.



### 3.2 Instance analysis of fog detection

Satellite data that is used for fog detection was obtained at 10:40 on January 30, 2009, covers several provinces in eastern China. Fig 15(a) is the true color images synthesized at band 1, 4 and 3 for RGB, green dots is the location of ground sites that observed the fog weather. Fig. 15 (b) is the cloud identification results. And Fig. 15 (c) is the

5   distribution of fog detected from the MODIS data of fig 15 (a), it indicates that for the heavy fog, most of the area can be accurately identified, but for the mist area and the edge area of fog distribution, like the areas of south of Anhui, Jiangsu provinces and north of Jiangxi and Zhejiang provinces in fig 15 (a), the recognition accuracy is poor.

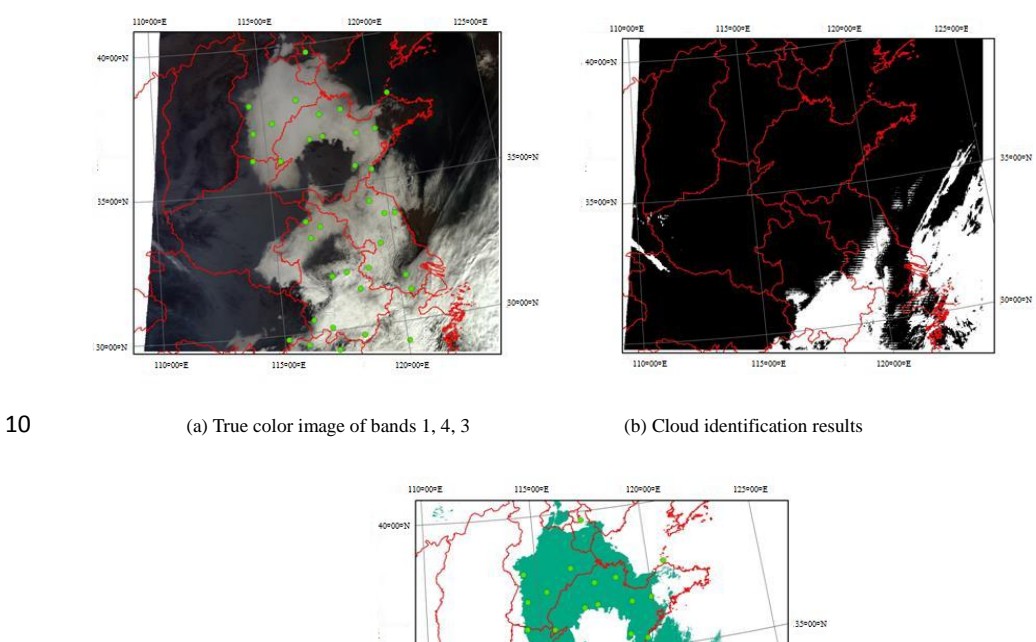

10              (a) True color image of bands 1, 4, 3                (b) Cloud identification results

(c) Fog detection results

**Fig 15.** The MODIS images obtained at 10:40 on 01.30.2009



Table 1 is the statistical results of the accuracy for fog detection, totally, 36 ground sites were distributed in the fog

area, among them, 9 sites were covered by clouds. As is shown in Table 1, the scope of the heavy fog at 9 sites

covered by cloud, as well 4 mist covered sites, were not detected. The detection accuracy of fog areas is 63.9%.

This accuracy might even higher if no cloud influence.

5     **Table 1.** The accuracy of fog detection

| Image acquired time | Observation time | Ground observation sites | Cloud-cove red sites | Identified sites | Invalid sites |
|---|---|---|---|---|---|
| 2009.01.30. 10:40 | 2009.01.30 11:00 | 36 | 9 | 23 | 13 |
| Percentage | | | 25% | 63.9% | 36.1% |
| Accuracy (including cloud influence) | | | 63.9% | | |
| Accuracy (excluding cloud influence) | | | 88.9% | | |

**3.3 Instance analysis of haze detection**

Fig. 16 (a) shows the MODIS data affected by haze weather at 14:00 on February 20, 2008. Green dots is the

location of ground haze observation sites, gray areas in the image is the haze distribution. It indicates that the haze

weather influenced a large area of central and eastern part of China. And a total of 85 ground haze detection sites

10     were distributed in such area, among them, 19 sites were covered by clouds, like the south of Hubei and Jiangxi

provinces and north of Fujian and Hunan provinces, which can be clear see in fig 16(a).

Fig. 16 (b) is the cloud identification and fig. 16(c) is the haze detection images. These images show that the haze

areas can be identified in high precision without cloud influence, however for cloud covered area, most of the haze

distribution can't be detected.



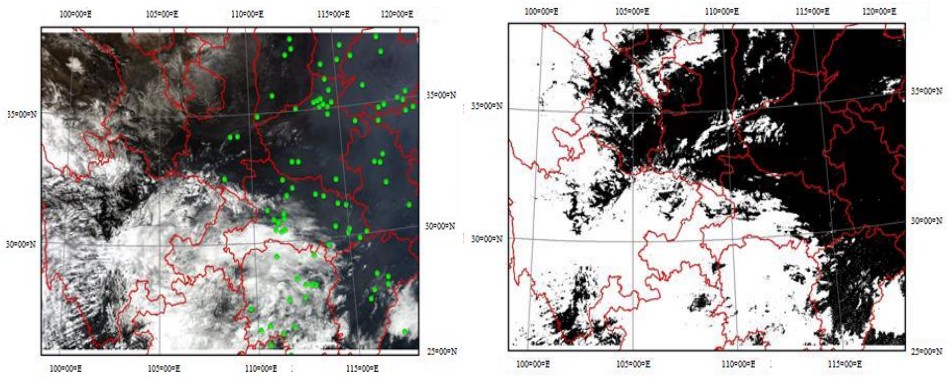

(a) True color image of bands 1, 4, 3          (b) Cloud identification results

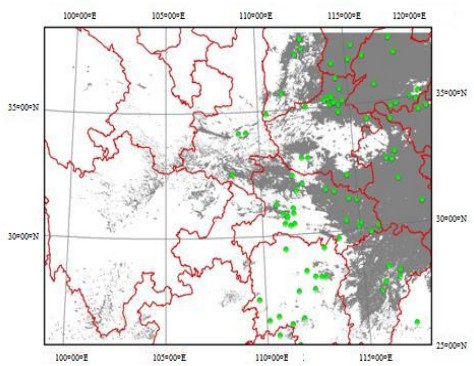

(c) Haze detection results

**5**          **Fig 16.** MODIS images obtained at 14:00 on 02.20.2008

As is shown in Table 2, 22.4% of all sites were covered by the cloud, with an extracting accuracy of 74.1%. This accuracy could even higher if without cloud influence.

Instance analysis indicates that the multichannel threshold value method proposed in this paper could have favorable effects on the detection of fog, haze and dust storm. However, due to cloud coverage, data from under the

**10**    cloud could not be detected by the MODIS sensor. Additionally, light fog areas were not detected, and the borders of haze and dust storm areas were detected poorly.



**Table 2.** The accuracy of haze detection

| Image acquired time | Observation time | Ground observation sites | Cloud-cove red sites | Identified sites | Invalid sites |
|---|---|---|---|---|---|
| 2008.02.20. 14:00 | 2008.02.20 14:00 | 85 | 19 | 63 | 22 |
| Percentage | | | 22.4% | 74.1% | 25.9% |
| Accuracy (including cloud influence) | | 74.1% | | | |
| Accuracy (excluding cloud influence) | | 96.5% | | | |

## 4 Conclusions

This paper uses MODIS as a main data resource in order to systematically analyze the physical and spectral characteristics of three types of extreme weather, namely, fog, haze and dust storms. The spectral differences between fog, haze, dust storm and clouds, as well as the underlying surface were analyzed, to determine the bands used for detecting the typical extreme weather events. Experiments of fog, haze, dust storm detection were performed based on the MODIS data, and the evaluations were made in conjunction with the data collected from MICAPS. The conclusions can be described as follows:

(1) When fog, haze or dust storm weather occur, dramatic changes occur in moisture and particle content; due to differences in their physical characteristics, the underlying surface and the three extreme weathers exhibit obvious differences in the behavior of their spectra. Remote sensing detection could be performed using the appropriate bands.

(2) In some wavebands, the three extreme weathers exhibit similar cloud reflectance values and brightness temperatures. Some cloud detection bands cannot be used to distinguish between clouds and the three weathers. Therefore, more bands are required for cloud detection in such weather condition.

(3) Experiments and validation shows that the detection methods for fog, haze and dust storm, developed in this research, can work well in most areas, but if they are covered by clouds, the accuracy is very poor, also this method is not good in the edge detection of the extreme weather.



**Acknowledgments**

The authors thank the Goddard Space Flight Center (GSFC) (http://ladsweb.nascom.nasa.gov) for providing the

MODIS data and the China Meteorological Administration for providing the MICAPS data. This work was supported

by the National Natural Science Foundation of China [41171270] and the Outstanding Youth Foundation of

Shandong Province [JQ201211].

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
