# Peer review of "A method for the spectral analysis and identification of Fog, Haze and Dust storm using MODIS data"

_Atmospheric Measurement Techniques, 2017_

## Referee Comment (RC1) · Anonymous Referee #3 · 17 Nov 2017

Review summary

In the present study it is described a methodology applied for the identification of various atmospheric scenes, over China, based on MODIS satellite data. To this aim, it is attempted a discrimination among fog, haze, dust, clouds and surface based on the spectral contrast of their brightness temperature or reflectance. The topic is interesting and fits to the research objectives of AMT. Nevertheless, there are many major drawbacks in the submitted manuscript which must be addressed before it can be accepted for publication. The authors must rewrite most parts of the text since the English writing is problematic making hard (in many cases even impossible) to understand the meaning. From a scientific point of view, I have doubts regarding the robustness of the obtained findings due to the limited number of cases that are analyzed. Different types

of clouds (depending on the meteorological conditions), the coverage of the underlying surface (depending on the season) or the intensity of fog, haze or dust storm may alter the results. All these aspects are not discussed in the manuscript. The interpretation of the scientific outcomes is poor without providing detailed explanations. Actually, the discussion is restricted just to a simple description of the figures. Moreover, in the results section there is not any reference whereas an intercomparison/connection between the obtained findings here and the corresponding ones reported from previous works is missing. Likewise, it would be useful to provide a map of the study region annotated with the locations mentioned in the text. In order to facilitate the visualization of your results, it would be better to keep the same line style for fog, haze, dust, clouds and surface among the histogram plots. Also, all the figures in which are depicted geographical distributions (e.g. Fig.1) must be improved. Please consider also to rewrite the captions in all figures/tables describing precisely what is presented there. Summarizing, I strongly believe that the submitted paper needs a major revision in order to be suitable for publication in AMT. Below are listed my comments/suggestions which I hope that will help the authors to improve their work.

Specific comments

1. Page 1, Lines 20-21: Don't use capital letter for Haze and Dust. The phrase "typical extreme weather" must be rephrased to "typical extreme weather conditions" or "typical extreme weather phenomena" or "severe weather conditions". I 've noticed that this incorrect statement is used throughout the manuscript. How the damage of the ecological environment is related to the development of fog?

2. Page 1, Lines 22-24: Modify this sentence because at its present form is not clear. I suppose that you want to point out that fog, haze and dust storm affect the radiation field within the Earth-Atmosphere system and these perturbations are related to impacts on climate.

3. Page 1, Lines 24-28: Merge these two sentences and reduce their length.

[Figure]

4. Page 2, Lines 1-2: Modify this sentence. Here is my suggestion: "For the identification of severe weather conditions several methods have been deployed relied on ground observations and spaceborne remote sensing techniques". Don't forget to add some references.

5. Page 2, Lines 2-6: Rewrite these two sentences highlighting in a better way the advantages/disadvantages of ground observations and satellite remote sensing techniques.

6. Page 2, Lines 6-8: "An accurate discrimination between fog, haze and dust is a challenging task due to their similar spectral signature influenced also by clouds which their properties can vary in a similar way in spectral terms".

7. Page 2, Lines 8-10: I cannot understand the meaning of this sentence. Please rephrase it.

8. Page 2, Lines 11-12: This sentence is not written appropriately.

9. Page 2, Lines 12 – 14: "For example, Eyre et al. (1984) used NOAA/AVHRR data to distinguish fog and low-level clouds based on differences of the brightness temperatures between channel 3 (3.7 $\mu$m) and channel 4 (11.0 $\mu$m).". When you are referring to previous works you have to add the relevant references as in my example. Please make the appropriate corrections in all relevant parts of the document. Is there any more recent reference that you can use?

10. Page 2, Lines 14-16: "Bendix et al. (2004), proposed a methodology suitable for the identification of fog during nighttime and daytime through a synergistic use of MODIS-Terra and MSG-SEVIRI retrievals."

11. Page 2, Lines 17-18: Is there any reference for that? Also, it would be useful to provide some numbers (e.g. frequency, concentrations, relative change expressed in percentages).

12. Page 2, Lines 17-21: The English writing must be improved. Moreover, it is not

so useful just to mention previous works without mentioning anything regarding the scientific findings.

13. Page 2, Lines 22-27: You have to provide more information in this part of the introduction. For example, which satellite data have been used by Shenk and Curram (1975)? Which dust storm Ackerman (1989) analyzed? The analysis made by Roskovensky and Liou (2005) focused on a specific dust outbreak? These are the only existing studies in the literature?

14. Page 3, Lines 1-6: Rephrase bullet 1. What do you mean "The remote sensing detection of haze is rarely performed...". I suppose that the authors want to highlight the low temporal resolution of satellite sensors sampling. Is that correct? Regarding the third bullet, if I have understood correctly, in the present analysis the fog, haze and dust are investigated for three individual cases. If so, then I cannot understand how the submitted work contributes towards this direction.

15. Page 3, Lines 7-9: "The optical and microphysical properties of fog, haze and dust particles vary among them, particularly in the visible and infrared wavelengths, attributed to differences in particles' sizes and moisture content.". Please add some references to support this statement.

16. Page 3, Lines 10-13: Reform this part of the text trying to make clear to the reader which is the overarching goal of your analysis.

17. Page 3, Section 2.1: The description of the utilized satellite data is very poor. For example, which data have been used? (Move all the information in this section and don't describe them in Section 2.2). From which satellite (Terra, Aqua or both) the retrievals have been acquired? Which version has been used? Where these data are available?

18. Page 3, Line 25: Correct the units.

19. Page 3, Line 26: Do you mean at the top of the atmosphere?

20. Page 4, Lines 4-5: "...E is the solar irradiance, D is the Earth-Sun distance (usually is equal to 1) whereas cos$\theta$ represents the cosine of the solar zenith angle."

21. Page 4, Lines 6-7: Provide the reference used for the definition of the brightness temperature.

22. Page 4, Lines 11-13: Replace "Boerziman constanst" with "Boltzmann constant". Provide units for the temperature.

23. Page 4, Lines 16-20: The authors should rewrite this part of the text. The information that they provide is vague and not so helpful for the reader. How these three different cases have been selected? A major drawback of the analysis is that they are studied only three cases reducing thus the robustness of the scientific outcomes. Is there any dependency of the defined thresholds, either for the brightness temperatures or for the surface reflectance, with respect to different atmospheric scenes? More specifically, how the different types of clouds, land cover or the intensity of the studied severe weather cases would affect (alter) the obtained results? Also, a map annotated with the locations stated in the manuscript will be useful since most of the readers are not familiar with the geography of the study region. Which is the meteorological department?

24. Page 6, Lines 1-5: The quality of the English writing is really bad. In addition, I don't think that this paragraph is useful. Each sentence can be used in the relevant parts of the document where Figures 1, 2 and 3 are discussed.

25. Page 6, Lines 6-10: Rewrite both sentences.

26. Page 6, Lines 10-12: The authors referring to conclusions without presenting any result so far. Moreover, could you support, based on previous relevant studies, how the bands have been selected?

27. Page 6, Lines 13-17: In Figure 4 is depicted the histogram of the brightness temperature differences between bands 22 and 23. Make the appropriate corrections

in the caption of Figure 4 adding also the label in x axis.

28. Page 6, Lines 17-18: "Thus, the fog area can be identified when T22-T23>9".

29. Page 7, Lines 1-2: Please add a reference.

30. Page 7, Line 5: "In Fig. 5 are illustrated the histograms of HI values...."

31. Page 7, Lines 6-8: Merge and rewrite these two sentences trying also to avoid the repetition of the Haze Index (HI). Replace "...identified if HI>0.03" with "...identified when HI>0.03".

32. Page 7, Line 11: In Figure 6 as well as in the other similar figures are presented histograms. Please consider to make the appropriate corrections throughout the manuscript.

33. Page 8, Lines 4-5: Remove this sentence and replace it with a better one in order to introduce the analysis that follows.

34. Page 8, Lines 6-9: Rewrite this part of the text. The sentence "In the brightness temperature difference image, differences in color represent differences in brightness temperatures" must be improved. It seems that the fog areas are depicted with gray colors, clouds with white-gray and the underlying surface with black. In Fig. 8, as in all relevant figures, are presented the histograms of temperature differences between specific bands. Please make the appropriate corrections. How can we distinguish the surface contribution since for brightness temperatures lower than 20 there is an overlapping area with the clouds' signal? It is not needed Figures 7 and 8 to be separated. Please merge also Figs. 9-10, 11-12 and 13-14.

35. Page 8, Lines 10-11: Can you explain better this sentence?

36. Page 8, Lines 11-12: "Based on the histogram of brightness temperature differences (Fig. ??) it is evident that the effect of the underlying surface effect can be eliminated when T20-T31>20."
37. Page 9, Lines 8-9: "Taking advantage of this spectral contrast, the differences of MODIS reflectances (b) between bands 19 and 3 are used for the identification of the underlying surface."

38. Page 9, Lines 11-13: The interpretation of both figures is just descriptive without providing any scientific aspect and also there is not any connection with previous works. For the identification of the surface contribution UI values must be higher than -0.1.

39. Page 10, Line 5: "In Fig. 11 is illustrated the TOA reflectances sampled by MODIS at band 1 while green circle symbols correspond to ground stations where dust storm has been recorded.". The interpretation of the results is poor (see comment 38).

40. Page 10, Line 8: I cannot understand this sentence.

41. Page 11, Lines 5-9: Rewrite this paragraph. Do you mean that clouds can be separated from land and dust when T20-T31 is lower than 24?

42. Page 12, Lines 7-9: Rewrite this part.

43. Page 12, Lines 9-10: You have to rephrase this sentence because it is not clear what are these 19 kinds of data available from the MICAPS software tool.

44. Page 13, Lines 2-7: You have to rewrite this part of the manuscript because the English writing is really bad. Moreover, there are some points which are unclear. For example, which satellite data have been used? I suppose that fog and cloud identification have been achieved based on figures 4 and 14, respectively. If so, then you can link parts of Section 3 with those presenting the applied methodology for the discrimination of different scene targets (Section 2).

45. Page 14, Lines 1-4: The discussion of the results presented in Table 1 must be improved. You have to add in the text a description of the information provided in the table. Can the authors comment the reasons leading to failure of their identification method applied for the detection of fog? Why do you have to include stations covered from clouds since it is impossible to identify fog from space-borne observations for these

cases? In the "Invalid sites" are considered also stations reporting cloudy/overcast conditions?

46. Page 14, Section 3.3: Rewrite this section. Please see also my previous comment.

47. Page 16, Conclusions: In the conclusions section you must to summarize the main findings of your work and not to refer to generic statements (e.g. different spectral variation between different sampled targets). Please consider to rewrite this section using a short introduction paragraph and then highlight the most important results.
* * *

---

## Referee Comment (RC2) · Anonymous Referee #1 · 23 Dec 2017

In this paper the authors selected three extreme events of fog, haze and dust storm and developed a methodology to distinguish between those three phenomena. Validation with ground data reveals that the method is promising. Overall, the main idea of the paper is interesting; however, the manuscript per se is not good and does not meet the standards of AMT. Therefore, I suggest to reject this paper due to the major issues mentioned below.

1. The paper suffers from language issues. At almost every sentence there is at least one grammatical error and at some points the manuscript is hard to follow. Unfortunately, at its current format the paper would not be accepted for publication not only in AMT but also in the majority of serious scientific journals.

2. The authors selected only three case studies which impairs the robustness of their

results. I have the feeling that the method works under the specific conditions for which it was developed but it is not sure if it can be used in other cases or on an operational basis. Land albedo changes from time to time and the method might not be applicable in other cases. What about mixed cases where fog and dust exist at the same time?

3. I disagree that dust storm should be considered as an extreme weather phenomenon. It is rather the result of specific weather types and synoptic transport than a weather phenomenon.

4. The quality of the images remains low and the captions are poor.

5. I have the feeling reading the paper that the method is not properly described. How did the authors decide to select the specific bands and indexes? Is there a theoretical basis? In that case there should be previous studies; however, there is not a single reference in the text.

---

## Author Comment (AC1) · 5 Feb 2018

We thank the referee for the constructive comments that we have tried to accommodate in the text. Detailed answers to the comments are given below (bold: referee comment, regular font: author's response).

**Review summary**

**In the present study it is described a methodology applied for the identification of various atmospheric scenes, over China, based on MODIS satellite data. To this aim, it is attempted a discrimination among fog, haze, dust, clouds and surface based on the spectral contrast of their brightness temperature or reflectance. The topic is interesting and fits to the research objectives of AMT. Nevertheless, there are many major drawbacks in the submitted manuscript which must be addressed before it can be accepted for publication. The authors must rewrite most parts of the text since the English writing is problematic making hard (in many cases even impossible) to understand the meaning. From a scientific point of view, I have doubts regarding the robustness of the obtained findings due to the limited number of cases that are analyzed. Different types of clouds (depending on the meteorological conditions), the coverage of the underlying surface (depending on the season) or the intensity of fog, haze or dust storm may alter the results. All these aspects are not discussed in the manuscript. The interpretation of the scientific outcomes is poor without providing detailed explanations. Actually, the discussion is restricted just to a simple description of the figures. Moreover, in the results section there is not any reference whereas an inter comparison/connection between the obtained findings here and the corresponding ones reported from previous works is missing. Likewise, it would be useful to provide a map of the study region annotated with the locations mentioned in the text. In order to facilitate the visualization of your results, it would be better to keep the same line style for fog, haze, dust, clouds and surface among the histogram plots. Also, all the figures in which are depicted geographical distributions (e.g. Fig.1) must be improved. Please consider also to rewrite the captions in all figures/tables describing precisely what is presented there. Summarizing, I strongly believe that the submitted paper needs a major revision in order to be suitable for publication in AMT. Below are listed my comments/suggestions which I hope that will help the authors to improve their work.**

**Answer:** Thank you very much for your recognition of our work, and also thank you for pointing out the problems in the article so that we can improve our work. In this revision, we have focused on the following aspects:

1) The English language is one of the biggest problems in our manuscript, and we are sorry for that. In order to improve the reading difficulties caused by the expression of language, we have commissioned a professional organization to help us polish the language, and we hope that we have got better improvements.

2) We add some content to illustrate the reasonableness, robustness, and limitations of our work. There are great difficulties in monitoring the three extreme weather phenomena mentioned in this paper by using remote sensing technology. At present, there are few methods of development, especially for the distinction between these three extreme weather phenomena. With the analysis of the spectral characteristics of three extreme weather phenomena, the extreme weather phenomena in the large area range are selected with the support of the surface observation data. The selected images include different surface types, different types of clouds, and the influence of different degrees of extreme weather. The above features can ensure the robustness of the method to a large extent. In addition, we have also increased the analysis of the difference in the material composition of the three extreme weather phenomena to make the reader clear the basis for the model construction.

3) We modified the content of the result analysis section and the content of the conclusion to highlight the work we have done. It is very regrettable that we have not found the same kind of work, so we have not increased the comparison with other's work.

4) A revision of the errors or irregularities of the language, chart, and formula mentioned by reviewers.

Thank you again for your work on this manuscript, and still welcome the new suggestion, so we can further improve it.

**Specific comments**

**1. Page 1, Lines 20-21: Don't use capital letter for Haze and Dust. The phrase "typical extreme weather" must be rephrased to "typical extreme weather conditions" or "typical extreme weather phenomena" or "severe weather conditions". I've noticed that this incorrect statement is used throughout the manuscript. How the damage of the ecological environment is related to the development of fog?**

**Answer:** Modified per your suggestion. The phrase "typical extreme weather" has been rephrased to "typical extreme weather phenomena" throughout the manuscript; The description of "the damage of the ecological environment is related to the development of fog" has been deleted.

**2. Page 1, Lines 22-24: Modify this sentence because at its present form is not clear. I suppose that you want to point out that fog, haze and dust storm affect the radiation field within the Earth-Atmosphere system and these perturbations are related to impacts on climate.**

**Answer:** Thank you. We modified this sentence as suggested. (**Page 1, Lines 24-25**)

**3. Page 1, Lines 24-28: Merge these two sentences and reduce their length.**

**Answer:** Done per your suggestion. (**Page 1, Lines 22-24**)

**4. Page 2, Lines 1-2: Modify this sentence. Here is my suggestion: "For the identification of severe weather conditions several methods have been deployed relied on ground observations and spaceborne remote sensing techniques". Don't forget to add some references.**

**Answer:** We modified this sentence as suggested and added some references. (**Page 1, Lines 27-28**)

**5. Page 2, Lines 2-6: Rewrite these two sentences highlighting in a better way the advantages/disadvantages of ground observations and satellite remote sensing techniques.**

**Answer:** We rewrited as suggested. (**Page 1, Lines 28-29; Page 2, Lines 1-4**)

**6. Page 2, Lines 6-8: "An accurate discrimination between fog, haze and dust is a challenging task due to their similar spectral signature influenced also by clouds which their properties can vary in a similar way in spectral terms".**

**Answer:** Dnoe. (**Page 2, Lines 5-7**)

**7. Page 2, Lines 8-10: I cannot understand the meaning of this sentence. Please rephrase it.**

**Answer:** We rephrased this sentence. (**Page 2, Lines 8-9**)

**8. Page 2, Lines 11-12: This sentence is not written appropriately.**

**Answer:** We have modified this sentence. (**Page 2, Lines 10-14**)

**9. Page 2, Lines 12 – 14: "For example, Eyre et al. (1984) used NOAA/AVHRR data to distinguish fog and low-level clouds based on differences of the brightness temperatures between channel 3 (3.7 m) and channel 4 (11.0 m)". When you are referring to previous works you have to add the relevant references as in my example. Please make the appropriate corrections in all relevant parts of the document. Is there any more recent reference that you can use?**

**Answer:** We corrected as suggested in all relevant parts of the document and added recent reference. (**Page 2, Lines 16-17**)

**10. Page 2, Lines 14-16: "Bendix et al. (2004), proposed a methodology suitable for the identification of fog during nighttime and daytime through a synergistic use of MODIS-Terra and MSG-SEVIRI retrievals."**

**Answer:** This sentence was modified to be "With the combined use of MODIS/Terra and MSG/SEVIRI data, Bendix et al. (2004) proposed a method for the identification of fog during the nighttime and daytime." (**Page 2, Lines 18-19**)

**11. Page 2, Lines 17-18: Is there any reference for that? Also, it would be useful**

to provide some numbers (e.g. frequency, concentrations, relative change expressed in percentages).

**Answer:** We rewrited as suggested. (**Page 2, Lines 25-27**)

**12. Page 2, Lines 17-21: The English writing must be improved. Moreover, it is not so useful just to mention previous works without mentioning anything regarding the scientific findings.**

**Answer:** We improved as suggested. (**Page 3, Lines 1-6**)

**13. Page 2, Lines 22-27: You have to provide more information in this part of the introduction. For example, which satellite data have been used by Shenk and Curram (1975)? Which dust storm Ackerman (1989) analyzed? The analysis made by Roskovensky and Liou (2005) focused on a specific dust outbreak? These are the only existing studies in the literature?**

**Answer:** Done per your suggestion. (**Page 3, Lines 7-19**)

**14. Page 3, Lines 1-6: Rephrase bullet 1. What do you mean "The remote sensing detection of haze is rarely performed ". I suppose that the authors want to highlight the low temporal resolution of satellite sensors sampling. Is that correct? Regarding the third bullet, if I have understood correctly, in the present analysis the fog, haze and dust are investigated for three individual cases. If so, then I cannot understand how the submitted work contributes towards this direction.**

**Answer:** We have modified the expression of these three contents. In the original expression, we want to explain the following two aspects: 1)The research work on haze detection is currently less; 2) The study of the distinction between three extreme weather phenomena is less. Now, we have made a revision to the main contribution of this article. (**Page 3, Lines 20-25**)

**15. Page 3, Lines 7-9: "The optical and microphysical properties of fog, haze and dust particles vary among them, particularly in the visible and infrared wavelengths, attributed to differences in particles' sizes and moisture content.". Please add some references to support this statement.**

**Answer:** We corrected as suggested and added some references to support this statement. (**Page 3, Lines 26-28**)

**16. Page 3, Lines 10-13: Reform this part of the text trying to make clear to the reader which is the overarching goal of your analysis.**

**Answer:** Done. (**Page 3, Line28; Page 4, Lines 1-2** )

**17. Page 3, Section 2.1: The description of the utilized satellite data is very poor. For example, which data have been used? (Move all the information in this section and don't describe them in Section 2.2). From which satellite (Terra, Aqua or both) the retrievals have been acquired? Which version has been used? Where these data are available?**

**Answer:** We merged Section 2.1 and Section 2.2, and increased data introduction. (**Page 4, Lines 5-15**)

**18. Page 3, Line 25: Correct the units.**

**Answer:** Done.

**19. Page 3, Line 26: Do you mean at the top of the atmosphere?**

**Answer:** Yes.

**20. Page 4, Lines 4-5: "E is the solar irradiance, D is the Earth-Sun distance (usually is equal to 1) whereas cos_ represents the cosine of the solar zenith angle."**

**Answer:** Done. (**Page 4, Lines18-20**)

**21. Page 4, Lines 6-7: Provide the reference used for the definition of the brightness temperature.**

**Answer:** We added the reference.

**22. Page 4, Lines 11-13: Replace "Boerziman constanst" with "Boltzmann constant". Provide units for the temperature.**

**Answer:** Done. (**Page 5, Lines2-5**)

**23. Page 4, Lines 16-20: The authors should rewrite this part of the text. The information that they provide is vague and not so helpful for the reader. How these three different cases have been selected? A major drawback of the analysis is that they are studied only three cases reducing thus the robustness of the scientific outcomes. Is there any dependency of the defined thresholds, either for the brightness temperatures or for the surface reflectance, with respect to different atmospheric scenes? More specifically, how the different types of clouds, land cover or the intensity of the studied severe weather cases would affect (alter) the obtained results? Also, a map annotated with the locations stated in the manuscript will be useful since most of the readers are not familiar with the geography of the study region. Which is the meteorological department?**

**Answer:** We have rewrite this part of the text. (**Page 5, Lines 6-20; Page 6, Lines 1-4**) The selected images include different surface types, different types of clouds, and the influence of different degrees of extreme weather. The above features can ensure the robustness of the method to a large extent. In addition, we have also increased the analysis of the difference in the material composition of the three extreme weather phenomena to make the reader clear the basis for the model construction. Also, a map annotated with the locations stated has been added. (**Page 6, Lines 9-10; Page 7, Lines 1-4**)

**24. Page 6, Lines 1-5: The quality of the English writing is really bad. In addition, I don't think that this paragraph is useful. Each sentence can be used in the relevant parts of the document where Figures 1, 2 and 3 are discussed.**

**Answer:** We have deleted this paragraph as your suggestion.

**25. Page 6, Lines 6-10: Rewrite both sentences.**

**Answer:** Done per your suggestion. (**Page 6, Lines 4-7**)

**26. Page 6, Lines 10-12: The authors referring to conclusions without presenting any result so far. Moreover, could you support, based on previous relevant studies, how the bands have been selected?**

**Answer:** We have added the physical characteristics of fog, haze and dust storms including composition, moisture content, particle size, color, border and diurnal variation to let the readers know why the bands were chosen. (**Page 5, Line10-20**)

**27. Page 6, Lines 13-17: In Figure 4 is depicted the histogram of the brightness temperature differences between bands 22 and 23. Make the appropriate corrections in the caption of Figure 4 adding also the label in x axis.**

**Answer:** We corrected as suggested. (**Page 7, Lines 5-9**)

**28. Page 6, Lines 17-18: "Thus, the fog area can be identified when T22-T23>9".**

**Answer:** We rewrited as suggested.

**29. Page 7, Lines 1-2: Please add a reference.**

**Answer:** We added as suggested.

**30. Page 7, Line 5: "In Fig. 5 are illustrated the histograms of HI values."**

**Answer:** Dnoe.

**31. Page 7, Lines 6-8: Merge and rewrite these two sentences trying also to avoid the repetition of the Haze Index (HI). Replace "identified if HI>0.03" with "identified when HI>0.03".**

**Answer:** We corrected as suggested.

**32. Page 7, Line 11: In Figure 6 as well as in the other similar figures are presented histograms. Please consider to make the appropriate corrections throughout the manuscript.**

**Answer:** We corrected as suggested in all relevant parts of the document. (**Page 8, Line 13**)

**33. Page 8, Lines 4-5: Remove this sentence and replace it with a better one in order to introduce the analysis that follows.**

**Answer:** Done per your suggestion. (**Page 9, Lines 5-8**)

**34. Page 8, Lines 6-9: Rewrite this part of the text. The sentence "In the brightness temperature difference image, differences in color represent differences in brightness temperatures" must be improved. It seems that the fog areas are depicted with gray colors, clouds with white-gray and the underlying**

**surface with black. In Fig. 8, as in all relevant figures, are presented the histograms of temperature differences between specific bands. Please make the appropriate corrections. How can we distinguish the surface contribution since for brightness temperatures lower than 20 there is an overlapping area with the clouds' signal? It is not needed Figures 7 and 8 to be separated. Please merge also Figs. 9-10, 11-12 and 13-14.**

Answer: Done per your suggestion. (**Page 9, Lines 9-16; Page 10, Lines 7-12; Page 10, Lines 13-17; Page 11, Lines 3-8**)

**35. Page 8, Lines 10-11: Can you explain better this sentence?**

Answer: We have rewrited this sentence. (**Page 9, Lines 10-12**)

**36. Page 8, Lines 11-12: "Based on the histogram of brightness temperature differences (Fig. ??) it is evident that the effect of the underlying surface effect can be eliminated when T20-T31>20."**

Answer: Done. (**Page 9, Lines 12-13**)

**37. Page 9, Lines 8-9: "Taking advantage of this spectral contrast, the differences of MODIS reflectances (b) between bands 19 and 3 are used for the identification of the underlying surface."**

Answer: Done. (**Page 10, Lines 1-5**)

**38. Page 9, Lines 11-13: The interpretation of both figures is just descriptive without providing any scientific aspect and also there is not any connection with previous works. For the identification of the surface contribution UI values must be higher than -0.1.**

Answer: We have changed the description of this content. (**Page 10, Lines 7-10**)

**39. Page 10, Line 5: "In Fig. 11 is illustrated the TOA reflectances sampled by MODIS at band 1 while green circle symbols correspond to ground stations where dust storm has been recorded." The interpretation of the results is poor (see comment 38).**

Answer: We corrected as suggested. (**Page 10, Lines 13-15**)

**40. Page 10, Line 8: I cannot understand this sentence.**

Answer: We reformed as follow: Thus, it is evident that most part of the underlying surface can be eliminated when b1>0.12. (**Page 10, Line 17**)

**41. Page 11, Lines 5-9: Rewrite this paragraph. Do you mean that clouds can be separated from land and dust when T20-T31 is lower than 24?**

Answer: We reformed as follow: Fig. 10a shows BTD image between bands 20 and 31, where the green dots represent ground-observed dust storm sites. Fig. 10b shows BTD histograms of dust storms, clouds and the underlying surface between bands 20 and 31. It shows that the underlying surface can be identified when $BT20-BT31$ is less

than 20. (**Page 11, Lines 3-5**)

**42. Page 12, Lines 7-9: Rewrite this part.**

**Answer:** We have rewrite this part (**Page 11, Lines 11-13**)

**43. Page 12, Lines 9-10: You have to rephrase this sentence because it is not clear what are these 19 kinds of data available from the MICAPS software tool.**

**Answer:** We have rewrited this paragragh. (**Page 11, Lines 13-16**)

**44. Page 13, Lines 2-7: You have to rewrite this part of the manuscript because the English writing is really bad. Moreover, there are some points which are unclear. For example, which satellite data have been used? I suppose that fog and cloud identification have been achieved based on figures 4 and 14, respectively. If so, then you can link parts of Section 3 with those presenting the applied methodology for the discrimination of different scene targets (Section 2).**

**Answer:** We have rewrited this paragragh. (**Page 12, Lines 1-9**)

**45. Page 14, Lines 1-4: The discussion of the results presented in Table 1 must be improved. You have to add in the text a description of the information provided in the table. Can the authors comment the reasons leading to failure of their identification method applied for the detection of fog? Why do you have to include stations covered from clouds since it is impossible to identify fog from space-borne observations for these cases? In the "Invalid sites" are considered also stations reporting cloudy/overcast conditions?**

**Answer:** Done per your suggestion. (**Page 13, Lines 1-6**)

**46. Page 14, Section 3.3: Rewrite this section. Please see also my previous comment.**

**Answer:** Done. (**Page 13, Lines 9-12**)

**47. Page 16, Conclusions: In the conclusions section you must to summarize the main findings of your work and not to refer to generic statements (e.g. different spectral variation between different sampled targets). Please consider to rewrite this section using a short introduction paragraph and then highlight the most important results.**

**Answer:** Done per your suggestion.   (**Page 15, Lines 6-24**)

---

## Author Comment (AC2) · 5 Feb 2018

We thank the referee for the constructive comments that we have tried to accommodate in the text. Detailed answers to the comments are given below (bold: referee comment, regular font: author's response).

**In this paper the authors selected three extreme events of fog, haze and dust storm and developed a methodology to distinguish between those three phenomena. Validation with ground data reveals that the method is promising. Overall, the main idea of the paper is interesting; however, the manuscript per se is not good and does not meet the standards of AMT. Therefore, I suggest to reject this paper due to the major issues mentioned below.**

**1. The paper suffers from language issues. At almost every sentence there is at least one grammatical error and at some points the manuscript is hard to follow. Unfortunately, at its current format the paper would not be accepted for publication not only in AMT but also in the majority of serious scientific journals.**

**Answer:** Thank you very much for your pertinent criticism on this article. We apologize for the problems in the previous work. We have made serious modifications to the problems you raised, for example, we have commissioned a professional organization to help us polish the language, seriously amended the nonstandard expression, modified the images with the quality problems, and we still very welcome you have more suggestions.

**2. The authors selected only three case studies which impairs the robustness of their results. I have the feeling that the method works under the specific conditions for which it was developed but it is not sure if it can be used in other cases or on an operational basis. Land albedo changes from time to time and the method might not be applicable in other cases. What about mixed cases where fog and dust exist at the same time?**

**Answer:** Thank you for pointing out possible problems in the robustness of methods. We have considered this problem, too. But because of the complexity of the effects of three extreme weather phenomena on radiation, it is difficult to determine the recognition algorithm by the radiative transfer equation. It is relatively simple to analyze the radiation difference between the three extreme weather phenomena and the spectral difference between them and the cloud or land surface. As you have mentioned, the surface type is complex, and the albedo of the surface is constantly

changing. It's hard to find enough cases to analyze all the possible problems. In order to ensure the robustness of the algorithm, here we have chosen a larger area for analysis with the support of surface observation data. Each study area has reached over hundreds of thousands of square kilometers. In this area, it contains basically all possible surface types, all possible types of clouds and extreme weather phenomena of varying degrees.

The purpose of our work is to achieve the distinction between different extreme weather phenomena. The construction of the algorithm is also an analysis of different extreme weather phenomena put together. Unfortunately, we have not found two or more extreme weather phenomena mixed cases. However, when we do the application, we do not know in advance what extreme weather conditions are. The algorithm can automatically identify what kind of extreme weather phenomenon exists in the image.

**3. I disagree that dust storm should be considered as an extreme weather phenomenon. It is rather the result of specific weather types and synoptic transport than a weather phenomenon.**

**Answer:** We agree with your definition of extreme weather. In this article, we mainly consider the strong effects of these three weather on the radiation and the impact on people and ecological environment.

**4. The quality of the images remains low and the captions are poor.**

**Answer:** Thank you for pointing out the problem of image quality. We have changed the image with the problems.

**5. I have the feeling reading the paper that the method is not properly described. How did the authors decide to select the specific bands and indexes? Is there a theoretical basis? In that case there should be previous studies; however, there is not a single reference in the text.**

**Answer:** Thank you for pointing out the problems we have in the description of the theoretical basis for method. We have increased the expression of the material composition and effects on the radiation of three extreme weather phenomena, which are the basis for our selection of band and index. (**Page 5, Lines 6-20**)